# Thouless pumping in Josephson junction arrays

Stavros Athanasiou, Ida E. Nielsen, Matteo M. Wauters and Michele Burrello$^{\star}$

Center for Quantum Devices and Niels Bohr International Academy, Niels Bohr Institute,
University of Copenhagen, DK–2100 Copenhagen, Denmark
$^{\star}$ michele.burrello@nbi.ku.dk

August 29, 2023

## Abstract

Recent advancements in fabrication techniques have enabled unprecedented clean interfaces and gate tunability in semiconductor-superconductor heterostructures. Inspired by these developments, we propose protocols to realize Thouless quantum pumping in electrically tunable Josephson junction arrays. We analyze, in particular, the implementation of the Rice-Mele and the Harper-Hofstadter pumping schemes, whose realization would validate these systems as flexible platforms for quantum simulations. We investigate numerically the long-time behavior of chains of controllable superconducting islands in the Coulomb-blockaded regime. Our findings provide new insights into the dynamics of periodically driven interacting systems and highlight the robustness of Thouless pumping with respect to boundary effects typical of superconducting circuits.

# 1 Introduction

Josephson junction arrays (JJAs) and their intricate many-body physics have captivated the attention of researchers since ground-breaking experiments in the 1990s (see the review [1]). Due to the possibility of engineering complex networks and their long-range coherence, JJAs have also been one of the leading candidates for quantum simulators in solid-state devices. Their practical application as quantum simulators, however, has been hindered by technological limitations: on one side, difficulties in tuning their physical parameters entailed the necessity of fabricating multiple devices to explore their phases of matter (see, for instance, Refs. [2–6]), thus impeding detailed investigations; on the other, irregularities in the self-capacitance, induced charge, and Josephson coupling of the superconducting elements, resulted in an uncontrolled disorder.

These limitations have been mitigated by recent advancements in epitaxial growth techniques [7] that have paved the way for the realization of clean superconductor-semiconductor (SC-SM) interfaces. These breakthroughs enable an unprecedented tunability of the Josephson couplings through electrostatic gates [8–12], as well as the fabrication of multiple quantum dots on the same hybrid device [13, 14], thereby revolutionizing the potential of JJAs as quantum simulators. Moreover, SC-SM platforms allow for on-chip patterning of arbitrary geometries in one and two dimensions, a precise control of the magnetic fluxes in these systems [11, 12], and a relatively easy scalability. All these elements motivate the theoretical design of novel phases of matter and quantum simulation protocols on controllable JJAs.

In this respect, topological phases of matter are an ideal target for quantum simulations in solid-state platforms due to their intrinsic robustness against disorder, noise, interaction, and possibly dissipation. When combined with a time-periodic driving, one can engineer novel out-of-equilibrium states with no static analogs, known as Floquet topological phases [15–20]. Among these, one of the simplest yet profoundly intriguing examples is Thouless pumping [21–23], a phenomenon arising in one-dimensional (1D) insulators with suitable time-periodic driving of the system parameters. The topology behind this phenomenon leads to quantization of the charge adiabatically pumped during each driving period; in 1D JJAs, this corresponds to a current $I = 2e\Omega\mathcal{C}$, with $\Omega$ being the pumping frequency and $\mathcal{C}$ a suitably defined Chern number characterizing the filled energy bands. Although theoretically well understood, experimental implementations of Thouless pumping have so far been confined to systems of ultracold atoms [24–27], optical waveguides [28], magneto-mechanical systems [29], and superconducting quantum processors [30] which fall short in capturing genuine transport phenomena of charged particles.

In this paper, we propose an innovative approach that combines a JJA with the ability to finely tune the induced charge on each SC island and the corresponding Josephson couplings. Through numerical simulations of 1D arrays in the Coulomb-blockaded regime, we focus on the long-time behavior of such systems. To investigate this limit, we connect a Josephson junction chain with superconducting leads that act as Cooper pair (CP) reservoirs. We study the effect of the electrostatic repulsion arising from the cross-capacitance between neighboring islands and show that topological pumping is remarkably robust with respect to both the coupling to the leads and nearest-neighbor interactions. Our proposal is qualitatively different from former experiments based on geometrical pumping which rely on optimal control of the pumping protocol (e.g. Ref. [31] for electron junction systems and Ref. [32, 33] for superconducting transistors). Topological pumping, on the other hand, is predicted to be robust against disorder [34–36] and imperfections in the modulations [22, 37]. Our findings shed new light on the role of interaction and dissipation in topological pumping schemes, which are currently at the core of intense debate [23, 38–40]. Through this research, we expand our understanding of JJAs as versatile platforms for quantum simulations while unveiling new insights into

topological phenomena and the dynamics of interacting systems.

The paper is organized as follows: In Sec. 2 we derive a bosonic model from the Hamiltonian of a 1D JJA. In particular, we specialize on two models characterized by nontrivial topological properties: the Rice-Mele (RM) model [41, 42], which we introduce in Sec. 2.1, and the Harper-Hofstadter (HH) model [43, 44], presented in Sec. 2.2. Our analysis of topological pumping in Josephson junction chains based on the RM model is presented in Sec. 3, where we investigate in detail the role of the coupling with external superconducting leads and the effects of nearest-neighbor interactions. In order to show that the findings are not model-specific, but hold on a wide class of periodically driven systems, we report further numerical analysis on the HH model in Sec. 4. In Sec. 5 we discuss the energy scales and constraints relevant to realistic experimental implementations. Finally, we draw our conclusions and present future outlooks in Sec. 6.

## 2 Josephson junction arrays in hybrid superconductor - semiconductor platforms

Recent developments in the epitaxial growth techniques of hybrid SC-SM materials [7] allow for the fabrication of 1D and 2D arrays of superconducting islands lithographically patterned in arbitrary geometries [10–12] and contacted to a semiconducting substrate through atomically pristine interfaces [45]. In such devices, the filling of the substrate can be controlled via a global electrostatic gate (a top gate in the experiments in Refs. [10–12]). Additionally, smaller gates can be used to locally change the potential and density of states of the SM [9].

In the following, we consider devices defined by a 1D chain of superconducting islands of sub-micrometer size. For Al islands, the typical critical temperature of such systems is $\sim$ 1.6K [10], and we regard the related superconducting gap as the largest energy scale in the description of these systems. As a consequence, when operating at temperatures of the order of a few tens of milliKelvins, as customary in experiments, we can neglect effects determined by the Bogoliubov quasiparticle excitations of the islands. We can thus describe the low-energy physics of these JJAs by considering solely the dynamics of their CPs.

For a neighboring pair of SC islands, the dynamics can be modeled through the interplay of two kinds of interaction; first, the electrostatic interaction determined by the capacitance matrix which describes not only the potential induced on one island by the charge of the other, but also the charges induced in both islands by the surrounding environment. Second, the tunneling of CPs between the two islands which define an effective SC-SM-SC junction where the coherent hopping of CPs is mediated by Andreev states [46]. These states are induced in the SM layer below the two islands, where superconductivity is induced by proximity [9], and in the in-between region. Both interactions can be affected by neighboring electrostatic gates. Let us consider, for instance, a 1D chain as the one depicted in Fig. 1. In this setup, each superconducting island is addressed by a side gate at potential $V_{g,j}$, controlling the induced charge $2en_{g,j} = V_{g,j}C_j^g$. Additionally, we assume that a cutter gate at potential $V_{c,j}$ can control the filling of the semiconducting region in proximity of each Josephson junction and thereby its transparency. In this way, the effective coherent hopping amplitude of CPs between the islands can be modulated by varying the carrier density in the SM [9] and potentially turned off by totally depleting the substrate.

At low temperature, the JJAs can be modeled through a standard quantum phase model (see, for instance, the review [1]). To describe a chain as the one depicted in Fig. 1, we assign a superconducting phase operator $\hat{\varphi}_j$ and a number operator $\hat{N}_j$ to each superconducting island. $\hat{N}_j$ defines the number of CPs in the island with respect to an arbitrary offset. The two operators obey the standard commutation relation $[\hat{N}_j, e^{i\hat{\varphi}_j}] = -e^{i\hat{\varphi}_j}$, and $e^{i\hat{\varphi}_j}$ annihilates a CP in the

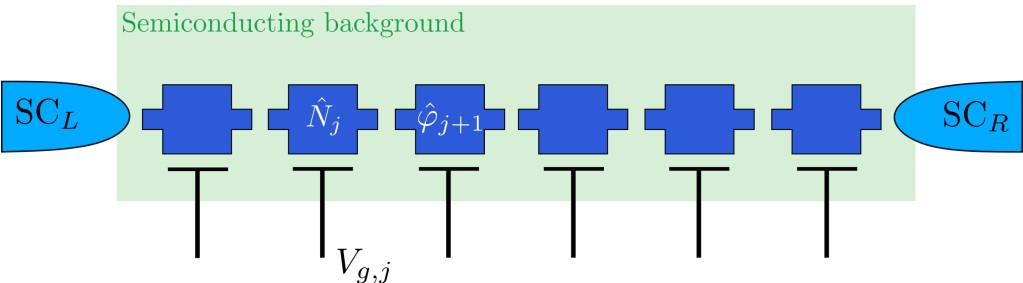

Figure 1: Chain of superconducting islands (dark blue) on top of a semiconducting substrate (green). Next to every island is a T-shaped side gate with tunable voltage $V_{g,j}$. At each end of the chain, there is a superconducting lead.

island $j$. The Hamiltonian for the chain is written as $\hat{H} = \hat{H}_C + \hat{H}_J$ and describes the interplay between the electrostatic interactions and CP tunneling, respectively. Let us first consider the charging energies of the islands in the chain, defined by

$$\hat{H}_C = 4e^2 \sum_{i,j=1}^{M} C_{ij}^{-1}(\hat{N}_i - n_{g,i})(\hat{N}_j - n_{g,j}). \tag{1}$$

Here $C^{-1}$ is the inverse capacitance matrix of the islands and $M$ is the number of islands in the chain. However, by assuming that the semiconducting environment and the electrostatic gates in the hybrid device effectively screen the charge of the SCs, we reduce the sum to single-island and nearest-neighbor interactions:

$$\hat{H}_C \approx E_C \sum_{j=1}^{M} (\hat{N}_j - n_{g,j})^2 + E_{CC} \sum_{j=1}^{M-1} (\hat{N}_j - n_{g,j})(\hat{N}_{j+1} - n_{g,j+1}). \tag{2}$$

In this equation, we introduced two customary energy scales: $E_C = 4e^2/C^{\text{self}}$ sets the charging energy of a single island, with $C^{\text{self}}$ being the sum of all capacitances to the other islands and environment elements; $E_{CC} = 4e^2(C^{-1})_{j,j+1}$ determines the electrostatic energy between neighboring islands. We will assume for simplicity that these quantities are translationally invariant, which is the expected behavior for a strong screening imposed by the environment. Weak variations along the chain, however, do not affect our analysis and can be easily accounted for. $n_{g,j}$ is the charge induced in each superconducting island and can be controlled by the surrounding electrostatic gates (see Fig. 1). In particular, we assume that each induced charge $n_{g,j}$ is primarily controlled by the voltage of the side gate addressing the island $j$, $n_{g,j} = V_{g,j} C_j^g/2e$. Here $C_j^g$ defines the mutual capacitance between the island $j$ and its neighboring side gate. More complex scenarios to account for the charge induced by all electrostatic gates can be easily investigated. The energy scale $E_C$ is determined by the geometry of the islands and their electrostatic environment. We consider, as an example, gated devices with Al islands patterned over an InAs 2D electron gas; for rectangular islands of size $\sim 750\text{nm} \times 80\text{nm}$, the resulting charging energy is approximately $E_C \approx 0.125\text{meV} \approx h30\text{GHz}$ [47].

The coherent tunneling of CPs is modeled by the Hamiltonian

$$\hat{H}_J = -\sum_{j=1}^{M-1} E_{J,j} \cos(\hat{\varphi}_{j+1} - \hat{\varphi}_j - \theta_{j,j+1}). \tag{3}$$

Here $E_{J,j}$ is the Josephson coupling between island $j$ and $j+1$ and the Peierls phase $\theta_{j,j+1} = \frac{2e}{\hbar c} \int_j^{j+1} \vec{A} \, d\vec{x}$ is the line integral of the vector potential $\vec{A}$ along a path between island $j$ and $j+1$,

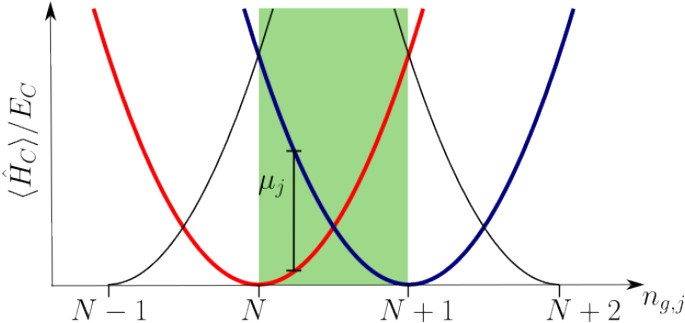

Figure 2: Charging energy of a single island as a function of the induced charge $n_{g,j}$ for states with $N-1$, $N$, etc. CPs. The thicker red and blue curves mark the two charge states we consider in the hardcore-boson approximation and the shaded area is the maximum region of validity of such an approximation. The effective chemical potential is the energy difference between the $N+1$ (blue) and $N$ (red) parabolas.

which accounts for the role of the magnetic field when embedding the chain in a closed superconducting loop. Eq. (3) corresponds to tunneling of single CPs between neighboring islands. Hence, we are neglecting terms characterized by higher harmonics of the phase difference $\hat{\varphi}_{j+1} - \hat{\varphi}_j$. In SC-SM-SC junctions, such an approximation is justified for cases in which the transmissibilities of the Andreev channels connecting the islands are low [46]. This, in turn, corresponds to a sufficiently depleted semiconducting substrate. Higher-harmonic terms can be considered as well, giving rise to non-quadratic interactions in terms of the annihilation and creation operators of CPs (for instance, coherent tunnelings of charge $4e$ objects). These are, however, considerably weaker than the energy scale $E_{J,j}$ of the single-CP tunneling, and we expect them not to play a crucial role in the implementation of Thouless pumping.

Concerning the Josephson amplitudes $E_{J,j}$, these can be globally controlled by a top gate as in Ref. [10], and the maximal value vastly depends on the width and length of the junction. Additionally, for the implementation of the RM model, we require the ability to separately address them through the voltage $V_{c,j}$ of suitable cutter gates. In general, the function $E_{J,j}(V_{c,j})$ may be complicated (see the experimental data related to gate-tunable devices in Refs. [48, 49]). We consider, however, a regime in which $E_{J,j} = 0$ when $V_{c,j}$ is below a certain threshold $V_{c,j}^*$, such that the substrate is totally depleted, and, for simplicity, we impose that $E_{J,j}$ is approximately linear in $V_{c,j}$ above this threshold [see Fig. 3(a)]. Importantly, the topological character of the RM model makes the pumped current robust against the details of the function $E_{J,j}(V_{c,j})$ as long as it is monotonic and sufficiently regular. Therefore, we consider a device in which the average value of the Josephson coupling $E_{J,j}$ is set by a global gate, whereas an additional periodic time modulation can be imposed by the cutter gate voltages. Experiments on hybrid Josephson junction showed that, for a width of about $0.3\mu$m, the amplitude $E_J$ at zero cutter voltage is of the order of $h50$GHz [9] and it can be switched off by applying sufficiently strong negative potentials (see, for instance, Ref. [49]).

When the Josephson energy dominates over the electrostatic terms, the system displays global phase coherence and behaves as a SC, allowing for coherent transport of CPs. Instead, the regime in which the electrostatic interaction $E_C$ dominates over the Josephson energies $E_{J,j}$, results in an insulating phase unless all the induced charges $n_{g,j}$ are fine-tuned close to $1/2$. In this scenario, the transport of CPs across the chain is suppressed, as we can consider each island in a Coulomb-blockaded state. In order to devise charge pumping protocols, we consider the regime $E_{C,j} > E_{J,j}$, $E_{CC,j}$, and, initially, we neglect the nearest-neighbor electrostatic interactions. We can rewrite the Hamiltonian $\hat{H}$ by introducing the operators $\hat{\Sigma}_j = e^{i\hat{\varphi}_j}$ and $\hat{\Sigma}_j^\dagger = e^{-i\hat{\varphi}_j}$ which, respectively, lower and raise the number of CPs in the island $j$ by 1,

such that $[\hat{N}_j, \hat{\Sigma}_j] = -\hat{\Sigma}_j$. We obtain:

$$\hat{H} = \sum_{j=1}^{M} E_C(\hat{N}_j - n_{g,j})^2 - \frac{1}{2} \sum_{j=1}^{M-1} E_{J,j} \left( \hat{\Sigma}^{\dagger}_{j+1} \hat{\Sigma}_j \, e^{-i\theta_{j,j+1}} + \hat{\Sigma}^{\dagger}_j \hat{\Sigma}_{j+1} \, e^{i\theta_{j,j+1}} \right). \tag{4}$$

This expression shows explicitly that the quantum phase model corresponds to a tight-binding Hamiltonian for the CPs.

The dispersion of the charging energy of a single island is depicted in Fig. 2 as a function of the induced charge $n_{g,j}$ for different number states. When $n_{g,j}$ is a half-integer, states that differ by one CP are degenerate; let us consider, for instance, the case $n_g \approx 0.5$. If $E_J \ll E_C$, we may assume that only the two lowest-energy states, with charge $N = 0$ and $N = 1$, are significantly occupied. States corresponding to the other parabolas are separated in energy by a gap $\sim 2E_C$ and their population in the many-body ground state is negligible. Therefore, under the assumption $E_C > E_{J,j}, T$, where $T$ is the system temperature, we can further simplify our description of the JJA and map the Hamiltonian $\hat{H}$ into a hardcore boson model. We may then replace $\hat{\Sigma}_j$ by a new hardcore boson operator $\hat{b}_j$, such that $\hat{b}^{\dagger}_j \hat{b}_j = \{0, 1\}$ and $\hat{b}^2_j = (\hat{b}^{\dagger}_j)^2 = 0$. The energy difference between the two lowest charge states is $E_C(1 - 2n_{g,j})$ for $n_g \approx 0.5$ (Fig. 2). Hence, we can define an on-site potential, resemblant of an effective chemical potential,

$$\mu_j = E_C(1 - 2n_{g,j}), \tag{5}$$

which vanishes for $n_{g,j} = 0.5$. We rewrite the total Hamiltonian (with $E_{CC} = 0$) as a tight-binding model of hardcore bosons:

$$\hat{H} = \hat{H}_C + \hat{H}_J \approx \sum_{j=1}^{M} \mu_j \hat{b}^{\dagger}_j \hat{b}_j - \frac{1}{2} \sum_{j=1}^{M-1} E_{J,j} \left( \hat{b}^{\dagger}_{j+1} \hat{b}_j \, e^{i\theta_{j,j+1}} + \text{H.c.} \right). \tag{6}$$

This Hamiltonian shows that, in the hardcore limit $E_C \gg E_{J,j}$, the system can be mapped into a chain of free fermions via a Jordan-Wigner transformation. Any time modulation of the electrostatic gates would translate into a time modulation of the onsite potentials $\mu_j$ and the tunneling amplitudes $E_{J,j}$. Hence, by implementing a suitable periodic modulation of these parameters, we expect that the Josephson junction chain is able to reproduce the physics of periodically driven systems of non-interacting fermions.

In the following, we will focus on the two most known schemes for the realization of adiabatic Thouless pumping: the periodically driven RM model and HH model. We restrict our analysis to the hardcore boson description but emphasize that the breakdown of this approximation and the effects of onsite interactions in the RM model have been theoretically investigated in Ref. [39].

Both models rely on a periodic drive of the onsite potential. We consider a generic time modulation of the side gates with

$$V_{g,j}(t) = V_{0,j} + \delta V_{g,j} \cos\left(\omega t + \chi_j\right), \tag{7}$$

where $\omega = 2\pi\Omega$. Such modulation yields:

$$\mu_j(t) = \mu_{0,j} - \delta\mu \cos\left(\omega t + \chi_j\right) \equiv E_C \left(1 - \frac{C^g_j V_{0,j}}{e}\right) - \frac{E_C C^g_j \delta V_{g,j}}{e} \cos\left(\omega t + \chi_j\right). \tag{8}$$

If $\mu_{0,j} \approx \bar{\mu}$ is approximately independent of the position along the chain, it plays the role of an overall chemical potential for the hardcore bosons. The static voltages $V_{0,j}$ can therefore be used to set the average filling of the system. In case of position-dependent variations of the

$C_j^g$ parameters, instead, these voltages can be tuned to reduce the fluctuations of the onsite potentials $\mu_{0,j}$ which, essentially, play the role of onsite disorder in the Hamiltonian (6). Finally, the oscillation amplitudes $\delta V_{g,j}$ determine the modulation of the onsite potential, which is additionally characterized by a position-dependent phase $\chi_j$ that we will suitably set to implement the RM and HH models, as described in the next subsections. Throughout this paper, we set $\hbar = 1$.

## 2.1   Rice-Mele Hamiltonian

The RM model [41] offers the most paradigmatic example of topological pumping in 1D systems. It is a model with a two-site unit cell, and its hardcore boson formulation is defined by a time-periodic Hamiltonian of the form

$$
\hat{H}_{RM}(t) = \sum_{j=1}^{M/2} \left( \mu_A(t)\hat{b}_{2j-1}^\dagger \hat{b}_{2j-1} + \mu_B(t)\hat{b}_{2j}^\dagger \hat{b}_{2j} \right)
$$

$$
- \frac{E_{J,1}(t)}{2} \sum_{j=1}^{M/2} \left( \hat{b}_{2j-1}^\dagger \hat{b}_{2j} + \text{H.c.} \right) - \frac{E_{J,2}(t)}{2} \sum_{j=1}^{M/2-1} \left( \hat{b}_{2j}^\dagger \hat{b}_{2j+1} + \text{H.c.} \right). \tag{9}
$$

Here we consider a chain with an even number of islands $M$ and open boundary conditions. The instantaneous spectrum of $\hat{H}_{RM}$ in the thermodynamic limit displays two bands with a linear level crossing at $E_{J,1} = E_{J,2}$ and $\mu_A = \mu_B$. For $\mu_A = -\mu_B$ the ground state is found at half-filling, and it is useful to define the two-component parameter vector $\vec{h} = \left( \frac{E_{J,1}-E_{J,2}}{2}, \mu_A - \mu_B \right)$. The time-dependent single-particle gap is given by $|\vec{h}(t)|$, and a topological charge pumping is obtained when $\vec{h}(t)$ winds around the gapless point $\vec{h} = 0$ during one period.

To implement Thouless pumping, we adopt a modulation of the kind in Eq. (8) for the onsite potentials and, in particular, we choose $V_{0,j}$ such that all islands are tuned close to the charge degeneracy point $\mu_{0,j} = 0$ between the lowest energy parabolas (Fig. 2). In order to enforce

$$
\mu_A(t) = -\mu_B(t) = \delta\mu \sin(\omega t), \tag{10}
$$

we set $\chi_j = (-1)^{j+1}\pi/2$ and choose modulation amplitudes $\delta V_{g,j}$ such that $\delta\mu$ is approximately constant along the chain. We observe that a residual $\bar{\mu}$ may reduce the energy gap of the system, but in the limit of adiabatic pumping, it does not affect the pumped charge as long as it remains sufficiently smaller than the single-particle gap at all times. The modulation of the Josephson energies $E_{J,j}(V_{c,j})$ is achieved by time-periodic voltages in the cutter gates. In particular, we adopt the following signals:

$$
V_{c,j}(t) = V_c + (-1)^{j+1}\delta V_c \cos(\omega t). \tag{11}
$$

We assume that all junctions in the chain are characterized by the same parameters, such that this modulation approximately results in Josephson amplitudes of the kind

$$
E_{J,j}(t) = f_{lr}\left[ J_0 + (-1)^{j+1}\delta J \cos \omega t \right], \tag{12}
$$

where $f_{lr}$ is a linear rectifier function, with $f_{lr}(x) = x$ if $x > 0$ and $f_{lr}(x) = 0$ otherwise, see Fig. 3(a). $V_c$ is used to control the offset $J_0$ and the modulation $\delta J$ is roughly proportional to $\delta V_c$. If $V_c - \delta V_c > V_c^*$, the Josephson amplitudes $E_{J,j}$ are always positive and display a sinusoidal modulation [see the example in Fig. 3(b)]. If instead $V_c - \delta V_c < V_c^*$, $E_{J,j}(t) = 0$ for a fraction of the time period and its modulation is clipped below zero [Fig. 3(c)].

The time-periodic nature of the Hamiltonian $H_{RM}$ allows us to consider the time coordinate as a second dimension for the momentum, such that we can assign a Chern number $\mathcal{C}$ to each

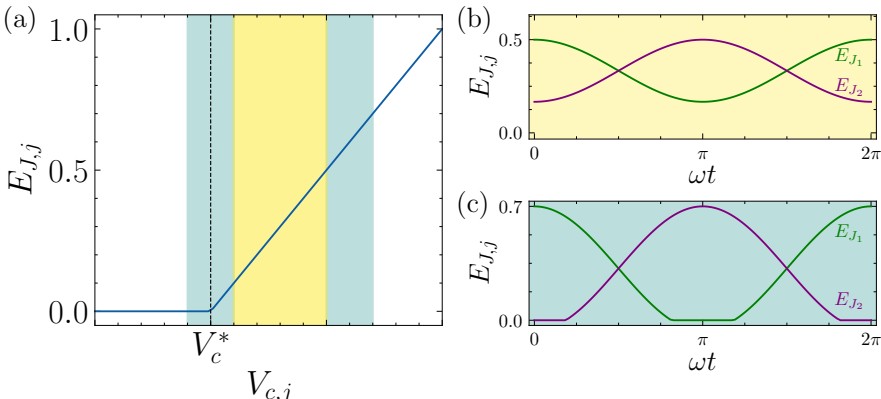

Figure 3: (a) The blue curve displays the approximation considered to model the Josephson energies $E_{J,j}$ as a function of the cutter gate voltage $V_{c,j}$ (arbitrary units). The dashed vertical line indicates the threshold $V_c^*$ below which $E_{J,j}$ is considered zero. Yellow and blue shaded areas schematically indicate the voltage range for the sinusoidal and clipped modulations, respectively. Panels (b) and (c) display the corresponding modulated $E_{J,j}(\omega t)$; green and purple curves refer to odd and even junctions, respectively.

energy band [21]. For the considered modulations, the Chern numbers are $\mathcal{C}_0 = 1$, $\mathcal{C}_1 = -1$, where the index $n = 0, 1$ labels the single-particle bands, resulting in an adiabatic Thouless pumping at half filling with average current $I = 2e\mathcal{C}_0\Omega$.

## 2.2 Harper-Hofstadter Hamiltonian

The 2D Hofstadter model [43] provides the simplest description of charged particles moving on a square lattice and subject to a uniform out-of-plane magnetic field. The dynamics of the model is determined by the magnetic flux per plaquette and the corresponding Aharonov-Bohm phase $\Phi$ acquired by a particle that moves around it. Incommensurate values of the flux result in the celebrated Hofstadter butterfly fractal spectrum, while for commensurate values $\Phi = 2\pi p/q$, the system can be modeled in the Landau gauge by introducing a $q$-site unit cell, and displays $q$ energy bands with non-trivial Chern numbers.

The equivalent driven 1D Hamiltonian, known as the Harper-Hofstadter model, is obtained by replacing one of the momentum components of the Hofstadter model by the time coordinate and, for hardcore bosons, it reads:

$$\hat{H}_{HH}(t) = \sum_{j=1}^{M} [\bar{\mu} - \delta\mu\cos(\omega t - \Phi j)]\,\hat{b}_j^\dagger\hat{b}_j - \frac{E_J}{2}\sum_{j=1}^{M-1}\left(\hat{b}_{j+1}^\dagger\hat{b}_j + \hat{b}_j^\dagger\hat{b}_{j+1}\right). \tag{13}$$

The realization of this model in the Josephson junction chains requires exclusively the modulation (8) of the onsite potentials and is obtained by setting a position-dependent phase $\chi_j = \Phi j$ in Eq. (7). In particular, setting $\Phi = 2\pi/3$ yields the simplest example of the gapped Hofstadter model. It corresponds to a 3-island periodicity of the phase $\chi_j$ and, in the thermodynamic limit, it yields three bands of the instantaneous spectrum (see Fig. 4), characterized by Chern numbers $\mathcal{C}_n = 1, -2, 1$. The band gap is proportional to the minimum between $E_J/2$ and $\delta\mu$.

The realization of the corresponding pumping scheme does not require a modulation of the Josephson amplitudes. Furthermore, the uniform and constant potential $\bar{\mu}$ can be changed by

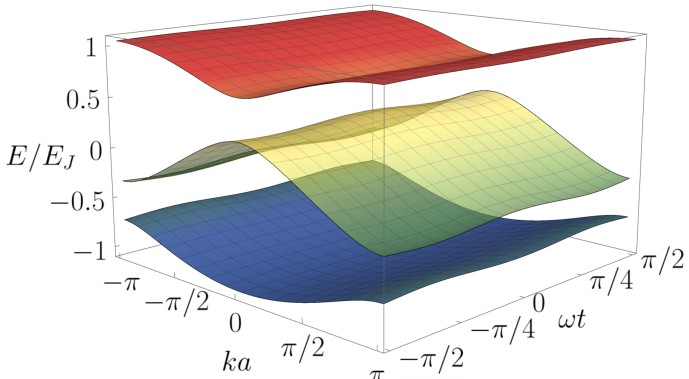

Figure 4: Spectrum of the Harper-Hofstadter model as a function of momentum and time at $\Phi = 2\pi/3$ for $\delta\mu = 0.4E_J$ and $\bar{\mu} = 0$.

regulating the voltage offsets $V_{0,j}$ and be used to set the overall filling of the CPs. The Thouless pumping is achieved when $\bar{\mu}$ lies in either of the band gaps depicted in Fig. 4. In particular, the charge pumped adiabatically in the thermodynamic limit is proportional to the sum of the Chern numbers of the filled bands: this implies that by varying $\bar{\mu}$ from the first to the second gap, we expect the pumped current to change sign from $2e\Omega$ to $-2e\Omega$.

## 2.3  Coupling with external superconducting leads

So far, we have discussed the modeling of isolated Josephson junction chains. To investigate their transport properties, however, it is necessary to embed these systems in a larger environment. Specifically, our aim is to model the driven transport of CPs across such JJAs. Therefore, the most convenient choice is to include two superconducting leads coupled to the first and last island of the chain (depicted in light blue in Fig. 1 and labeled by $SC_L$ and $SC_R$). To this purpose, we supplement the Hamiltonian $\hat{H}$ with a boundary term that describes a Josephson junction between the two extreme islands and the related superconducting leads [50]

$$\hat{H}_b = -E_L \left[ \cos\left(\hat{\varphi}_1 - \hat{\varphi}_L\right) + \cos\left(\hat{\varphi}_R - \hat{\varphi}_M\right)\right], \tag{14}$$

where $E_L$ describes the associated Josephson energy which we consider constant in time. By assuming that both leads are described by standard BCS coherent states, we can replace the related phase operators $\hat{\varphi}_{L/R}$ with classical phases $\varphi_{L/R}$. When embedding the JJA in an external SQUID, the phase difference $\phi = \varphi_R - \varphi_L$ would correspond to the magnetic flux threading it. We observe that such a phase difference can be absorbed in the Peierls phases $\theta_{j,j+1}$ in Eq. (3) by a suitable gauge transformation.

In the hardcore limit, the boundary Hamiltonian $\hat{H}_b$ becomes

$$\hat{H}_b = -\frac{E_L}{2} \left[ e^{i\varphi_L} b_1^\dagger + e^{i\varphi_R} b_M^\dagger \right] + \text{H.c.}. \tag{15}$$

This boundary term violates the conservation of the total number of CPs in the chain, such that, in general, the many-body ground and Floquet states will be superpositions of different particle numbers. Hence, the adiabatic condition no longer involves the single-particle band gap $e_g$ but a *many-body energy gap* $E_g$ that depends on the interplay between the charging energies of the SC islands, the Josephson energies of the junctions, and the coupling with the leads.

## 3 Results of the Rice-Mele model

In the following, we will investigate Thouless pumping in short chains coupled to external SC leads to identify experimentally relevant regimes where charge quantization can be observed. In particular, we will focus on the role played by the Josephson coupling to the leads, which breaks particle number conservation, and on the effects of the nearest-neighbor interaction $E_{CC}$. To this end, we adopt a Floquet description of the driven Josephson chain in the hardcore boson limit. We consider the Hamiltonian

$$\hat{H}_{RM,\text{tot}}(t,\phi) = \hat{H}_{RM}(t) + \hat{H}_b(\phi), \tag{16}$$

which describes a RM chain, as defined in Eq. (9), embedded in a superconducting circuit through the contact with the leads in Eq. (15). $\hat{H}_{RM,\text{tot}}(t,\phi)$ is periodic in both $t$ and $\phi$, and we can analyze the dynamics of the related system by defining a many-body Floquet operator which, for a generic Hamiltonian $\hat{H}$, reads

$$U(\tau) = \mathcal{T} e^{-i \int_0^\tau \hat{H}(t)\, \mathrm{d}t}. \tag{17}$$

Here $\mathcal{T}$ indicates time ordering and $\tau = 1/\Omega$ is the time period such that $\hat{H}(t+\tau) = \hat{H}(t)$. We consider system sizes up to $M = 8$, compatible with realistic devices in which all islands can be independently addressed by gate voltages, and we numerically diagonalize $U(\tau)$ to obtain the many-body Floquet eigenstates $\{|\Phi_\nu(\tau)\rangle\}$:

$$U(\tau)|\Phi_\nu(\tau)\rangle = e^{-i\mathcal{E}_\nu \tau}|\Phi_\nu(\tau)\rangle, \tag{18}$$

where $\mathcal{E}_\nu$ labels the many-body Floquet quasienergies. In the infinite-time limit, for a given phase difference $\phi$, the time-averaged pumped charge per cycle corresponds to [22, 51, 52]

$$Q_\infty \equiv \lim_{m \to \infty} \frac{2e}{m} \int_0^{m\tau} \mathrm{d}t'\, \mathrm{tr}\left[\rho(t')\partial_\phi \hat{H}(t',\phi)\right] = 2e\tau \sum_\nu \mathcal{N}_\nu(\phi)\, \partial_\phi \mathcal{E}_\nu, \tag{19}$$

with

$$\mathcal{N}_\nu(\phi) = |\langle \Phi_\nu(\tau)|\Psi_0\rangle|^2. \tag{20}$$

Here $\Psi_0$ is the state of the Josephson chain at the beginning of the pumping protocol, which we set to the ground state of $\hat{H}_{RM,\text{tot}}(t=0,\phi)$, $\rho(t)$ is the density matrix of the system at time $t$, and $2e\partial_\phi \hat{H} = \hat{\mathcal{J}}$ corresponds to the current operator. In the adiabatic limit, the ground state $|\Psi_0\rangle$ of $\hat{H}_{RM,\text{tot}}(t=0,\phi)$ has a large overlap with the Floquet state with the lowest energy expectation value [44], which we denote by $|\Phi_0(\tau)\rangle$, such that $\mathcal{N}_\nu \to \delta_{\nu,0}$. Hence, $Q_\infty$ is carried by a single Floquet state. An equivalent expression for the pumped charge per cycle in the infinite-time limit is [52]

$$Q_\infty = \sum_\nu \mathcal{N}_\nu(\phi) \int_0^\tau \mathrm{d}t \langle \Phi_\nu(t)|\hat{\mathcal{J}}(t)|\Phi_\nu(t)\rangle, \tag{21}$$

where $\hat{\mathcal{J}}(t)$ is the time-dependent current operator, and $|\Phi_\nu(t)\rangle$ are the time-evolved Floquet states *within one period*. Given the difficulty of accurately differentiating numerically the quasienergies with respect to the external SC phase, due to the many (avoided) level crossings in the Floquet spectrum, we will use Eq. (21) to extract the pumped charge from the time evolution. Our numerical calculations are performed with the QuTiP Python framework [53, 54].

The phase difference between the two external superconducting leads, rescaled by the number of islands in the chain, $\phi/M$, can be interpreted as a shift of the quasimomenta of

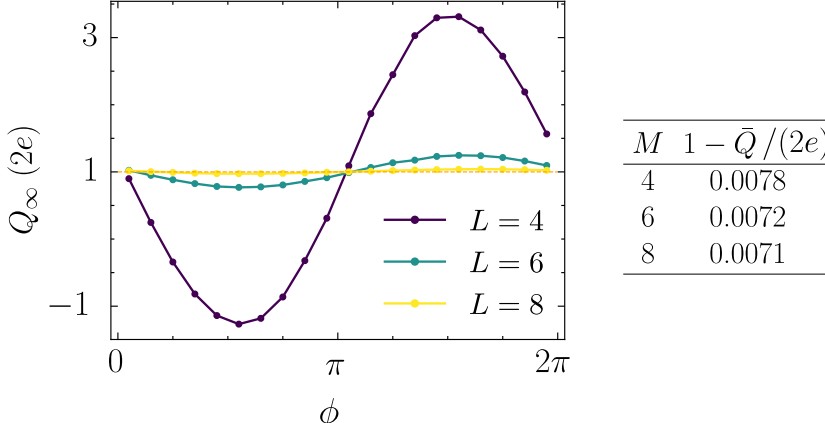

Figure 5: Pumped charge at infinite time $Q_\infty$ as a function of the phase difference $\phi$ between the external superconducting leads for $M = 4, 6, 8$ islands. The data correspond to the following parameters: $\delta J = 0.7 J_0, \delta\mu = 1.5 J_0, E_L = 2 J_0, \omega = 0.05 J_0$. The table represents the deviations from perfect quantization of the phase averages pumped charge $\bar{Q}$ as a function of the systems size $M$.

the system CPs via a suitable gauge transformation. For small systems, this shift causes a nontrivial dependence of $\partial_\phi \mathcal{E}_\nu$ on $\phi$ that yields, in turn, a dependence of $Q_\infty$ on the external phase. This is expected to be relevant only for small systems and vanishes completely in the thermodynamic limit, provided the system is in an insulating phase [55]. To evaluate these finite-size effects for the Hamiltonian $\hat{H}_{RM,\text{tot}}$, we depict in Fig. 5 the time-averaged pumped charge $Q_\infty(\phi)$ for $E_{CC} = 0$, $E_L = 2 J_0$ and a ratio $\sim 1/16$ between $\omega$ and the *many-body* gap $E_g$. For $M = 4$ and $M = 6$ islands, the dependence on $\phi$ is significant, but the oscillation around the average quantized value reduces to $\sim 2\%$ for $M = 8$. We find analogous results for the HH model, see Sec. 4.

To recover an exact quantization in short chains, it is therefore necessary to average the pumped charge over $\phi$. By following the Thouless construction [34], we define

$$\bar{Q} = 2e\tau \sum_\nu \int_0^{2\pi} \frac{d\phi}{2\pi} \mathcal{N}_\nu(\phi)\, \partial_\phi \mathcal{E}_\nu = \int_0^{2\pi} \frac{\mathrm{d}\phi}{2\pi} Q_\infty(\phi). \tag{22}$$

For systems with periodic boundary conditions conserving the particle number (such that $\phi$ is a phase twist in the boundary conditions), this quantity is quantized in the adiabatic limit at half-filling and corresponds, when $\mathcal{N}_\nu = \delta_{\nu,0}$, to the Chern number of the lowest energy band. More in general, Eq. (22) implies that only Floquet many-body states whith a quasienergy that winds in the Floquet-Brillouin zone as a function of $\phi$ contribute to the pumped charge $\bar{Q}$ (see, for instance, Fig. 7). Our results show that $\bar{Q}$ displays a precise quantization towards the adiabatic limit within an error of 1%, even when including the boundary term $\hat{H}_b$. The residual small deviation originates from non-adiabatic effects due to the finite frequency and the choice of the driving protocol [42, 44]. Adiabatic perturbation theory gives a leading-order estimate of this correction which scales as $1 - \bar{Q} \sim \left(\frac{\omega}{E_g}\right)^2$.

In the following, we will consider small system sizes, for which the phase dependence is sizable, and we will investigate the dependence of the pumped charge on the modulation $\delta J$ of the Josephson coupling, the coupling $E_L$, and the nearest-neighbor interaction $E_{CC}$.

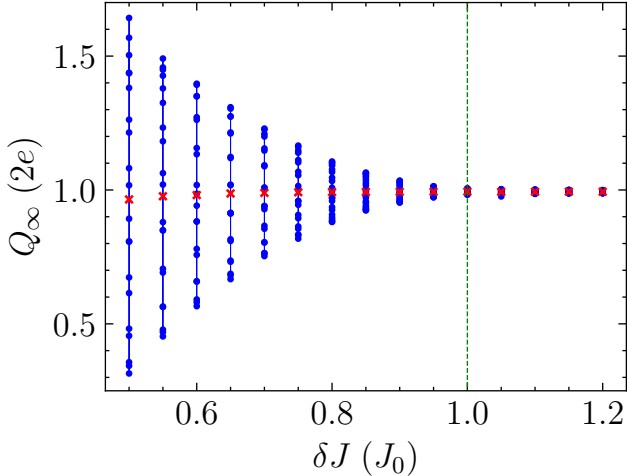

Figure 6: Pumped charge $Q_\infty$ of the RM model for 20 discretized values of $\phi$ (blue points) as a function of the Josephson energy modulation $\delta J$. The red crosses correspond to $\bar{Q}$. The phase dependence and the charge quantization considerably improve approaching the clipped modulation. The driving frequency is $\omega = 0.05J_0$.

## 3.1 Role of the Josephson modulation

The amplitude $\delta V_c$ of the cutter gate modulation and, consequently, the amplitude $\delta J$ in Eq. (12), determine the waveform of $E_{J,j}(t)$ (see the examples in Fig. 3). In this respect, the role of $\delta J$ is twofold: on one hand, it determines the minimum single-particle bulk gap $\epsilon_g = \min\left(|\vec{h}(t)|\right)$ over one period, which is given by $\min\left(\delta J, \frac{J_0 + \delta J}{2}, 2\delta\mu\right)$. Consequently, $\delta J$ plays a major role in determining non-adiabatic corrections to the pumped charge quantization. On the other hand, $\delta J$ determines whether the chain dimerizes exactly during the driving protocol. Although this has no effect in the thermodynamic limit, as the topological nature of Thouless pumping makes it insensitive to such details, it becomes important for the short chains we consider in this work.

To estimate these effects, we consider the dependence of $Q_\infty$ on $\delta J$ for different values of $\phi$ (Fig. 6) in an array with $M = 6$ islands. In the sinusoidal regime, where $\delta J$ is small, finite-size effects induce a large dispersion of $Q_\infty$ as a function of $\phi$, and averaging over the phase is fundamental to obtain the charge quantization. The dispersion rapidly decreases by increasing $\delta J$, until the variation of $Q_\infty(\phi)$ drops below 0.8% when $\delta J \geq J_0$. The weak dependence of $\bar{Q}$ (red crosses) on $\delta J$, instead, originates from non-adiabatic corrections. These are suppressed with $\delta J$ since the band gap $\epsilon_g$ increases accordingly, bringing the system deeper into the adiabatic regime without changing the driving frequency. In particular, the residual nonadiabatic and finite-size corrections of $\bar{Q}$ are less than 0.6% for $\omega = 0.05J_0$ in the clipped regime.

This dependence on the Josephson energy modulation can be understood by comparing the behavior of populated Floquet many-body states as a function of $\phi$ for $\delta J = 0.5J_0$ (sinusoidal waveform) and $\delta J = 1.2J_0$ (clipped waveform). They are shown in the left and right panels of Fig. 7, respectively, where we plot the Floquet quasienergies as a function of the phase $\phi$, using the corresponding occupation number $\mathcal{N}_\nu$ to set the color of each data point. In both cases, a single low-energy Floquet state $|\Phi_0(\tau)\rangle$ has an almost perfect overlap with the initial ground state, allowing us to follow the winding of $\mathcal{E}_0$ around the Floquet-Brillouin zone and confirming the quantization of $\bar{Q}$, which follows from Eq. (19). In the sinusoidal regime, $\mathcal{E}_0$ is, in general, far from linear in $\phi$, leading to a strong dependence of $Q_\infty(\phi)$ on this phase. In the clipped regime, instead, the derivative $\partial_\phi \mathcal{E}_0$ of the most populated state is practically constant,

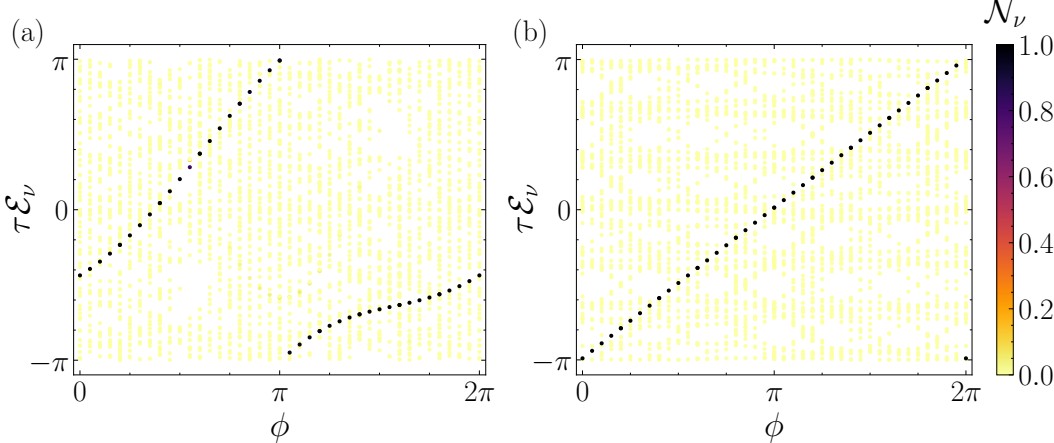

Figure 7: Comparison of the many-body Floquet eigenenergies of the RM model for (a) small sinusoidal modulations $\delta J = 0.5 J_0$ (b) and clipped modulation $\delta J = 1.2 J_0$ of the Josephson tunnelings. The dot colors represent the overlap $\mathcal{N}_\nu(\phi)$ of the initial ground state with the Floquet states. In the clipped regime $\partial_\phi \mathcal{E}_0$ is almost constant, leading to a pumped charge almost independent on the phase. The data refer to a system with $M = 6$ islands, $\omega = 0.05 J_0$, and $E_L = 2 J_0$.

and, for the chosen value of $\omega$, $\mathcal{N}_0 \sim 1 - O(\omega^2 / E_g^2)$ for all values of $\phi$. We conclude that even with small systems ($M = 6$), the phase dependence is very weak in the clipped regime, and a good quantization can be observed without averaging over the SC phase difference.

## 3.2 Coupling with the leads

Besides the dependence of the pumped charge on the external phase $\phi$, it is important to investigate the role of the coupling $E_L$ between the edge islands and the superconducting leads. When $E_L$ dominates over the other energy scales, the first and last islands are effectively merged with the two external leads and, in practice, the system behaves as a shorter chain. If instead $E_L$ is weak compared to $J_0$, the transport of CPs across the system is hindered, as the pumped current is faster than the rate of charge transferring to or from the leads. Consequently, the system behaves as an open chain. To examine the interpolation between these two limits, it is instructive to consider the behavior of the system for a clipped modulation $\delta J > J_0$. In this situation, the onsite potentials $\mu_{A/B}$ and $E_{J,1}$ vanish in the middle of the pumping period, $t = (n + 1/2)\tau$. Therefore, the ground state of the Hamiltonian $\hat{H}_{RM}(\tau/2)$ corresponds to the extreme topological dimerized state of the Su-Schrieffer-Heeger (SSH) model, and it displays the typical four-fold degeneracy associated with localized zero-energy boundary modes. The boundary Hamiltonian $\hat{H}_b$, however, gaps the SSH edges; the ground states of the first and last islands in this dimerized limit result

$$\left| \psi_{1/M}(\tau/2) \right\rangle = \frac{1}{\sqrt{2}} \left( \left| \hat{N}_{1/M} = 0 \right\rangle + e^{i\varphi_{L/R}} \left| \hat{N}_{1/M} = 1 \right\rangle \right) \tag{23}$$

and are separated by an energy gap $E_L$ from the localized excited states in the same islands. Therefore, when $E_L < \frac{J_0 + \delta J}{2}, 2\delta\mu$, it sets the many-body energy gap $E_g$. Notably, the limit $E_L \to 0$ corresponds to the isolated RM chain in which the zero-energy SSH edge modes cause transitions between the two energy bands, thereby disrupting the Thouless pumping. Only when $E_L$ is sufficiently larger than $\omega$, this non-adiabatic effect vanishes and the quantized pumping can be restored.

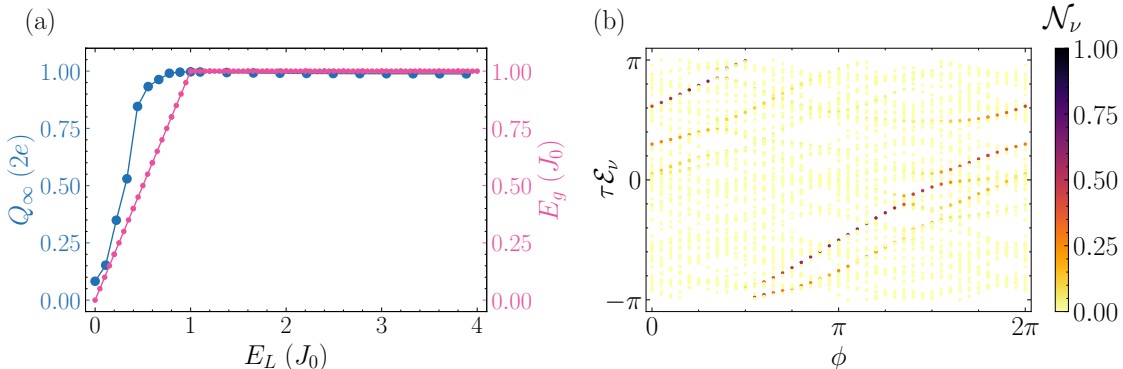

Figure 8: (a) Pumped charge $Q_\infty(\phi = 0)$ and many-body energy gap $E_g$ as a function of $E_L$ for $\delta J = J_0$ (threshold between clipped and sinusoidal modulations). The data are obtained in a system with parameters $\delta\mu = 1.5J_0$, $\omega = 0.05J_0$, and $M = 6$. (b) The many-body Floquet eigenenergies of the Rice-Mele model for $\delta J = J_0$ and $E_L = 0.5J_0$. In this situation, multiple Floquet states overlap significantly with the initial ground state resulting in a non-quantized current. Data are obtained for 6 islands and a driving frequency of $\omega = 0.05J_0$.

Fig. 8(a) displays the pumped charge $Q_\infty(\phi = 0)$ and the many-body gap $E_g$ as a function of $E_L/J_0$ for a modulation $\delta J = J_0$ and a driving frequency $\omega = 0.05J_0$. The two quantities are clearly correlated; $Q_\infty$ saturates to the quantized value as the gap induced by the coupling with the leads becomes sufficiently large. As expected, $E_g$ eventually saturates as well at the value $E_g = J_0$, when $E_L \geq J_0$. In the case of sinusoidal modulation, instead, the dimerized SSH limit is never reached during the driving period. Therefore the many-body gap saturates at smaller values for large $E_L/J_0$.

The region $0 < E_L < J_0$ is the most susceptible to non-adiabatic errors since the ratio between the many-body gap and the driving frequency changes continuously between 0 and $J_0/\omega$. Hence, we expect the pumped charge to saturate faster to $Q_\infty = 2e$ for smaller values of $\omega$. In the adiabatic limit $\omega \to 0$, a finite gap opens for any value of $E_L > 0$, leading to the quantization of $Q_\infty$. However, in a realistic scenario, the driving frequency is finite and sets the average magnitude of the current flowing through the array. Thus, $E_L$ needs to be large enough to prevent the CPs from accumulating on one side of the chain, similar to what happens in an open system, suppressing the charge pumping. In the Floquet framework, this can be understood by observing the winding of the quasienergies associated with Floquet states with the highest occupation. When the charge is quantized and the driving is sufficiently slow, there is a clear correspondence between Floquet states and energy eigenstates, leading to an almost perfect fidelity $\mathcal{N}_0 \simeq 1$, for any value of the phase difference $\phi$. Instead, when $E_L < J_0$, there are multiple Floquet states that display a large overlap with the initial ground state, as shown in Fig. 8(b), which might interfere destructively. Moreover, large avoided crossings appear in the spectrum as $\phi$ changes, further suppressing quantized transport.

We conclude that, in order to avoid additional non-adiabatic corrections to the pumped charge, $E_L$ must be larger than the single-particle band gap of the chain.

## 3.3   Nearest-neighbor interactions

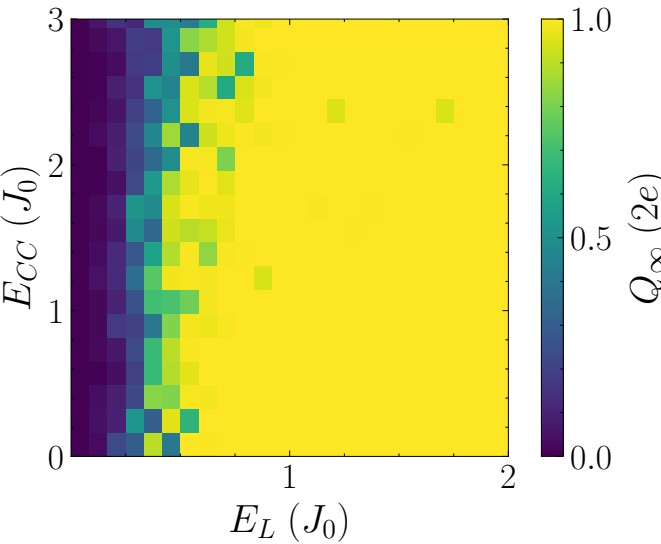

Figure 9: Pumped charge $Q_\infty(\phi = 0)$ as a function of the coupling with the leads $E_L$ and the nearest neighbor interactions $E_{CC}$ for the RM model with 6 islands. The data are obtained with $\delta J = J_0$, $\omega = 0.05 J_0$ and $\delta\mu = 1.5 J_0$.

Differently from the most common trapped ultracold atom systems [23], JJAs can be characterized by sizeable nearest-neighbor interactions. These interactions are bounded by $E_{CC} < E_C$, however, in the hardcore limit, they can be highly relevant because nothing prevents $E_{CC}$ from being of the same order of $J_0$ or the single-particle gap. For hardcore bosons, the nearest-neighbor interaction reads:

$$\hat{H}_{\text{int}}(t) = E_{CC} \sum_{j=1}^{M-1} \left(\hat{b}_j^\dagger \hat{b}_j - n_{g,j}(t)\right)\left(\hat{b}_{j+1}^\dagger \hat{b}_{j+1} - n_{g,j+1}(t)\right). \tag{24}$$

$\hat{H}_{\text{int}}$ has two effects on the chain: a static repulsion, $E_{CC}\hat{b}_{j+1}^\dagger \hat{b}_j^\dagger \hat{b}_{j+1}\hat{b}_j$, and a shift in the chemical potential, $-E_{CC}(n_{g,j-1} + n_{g,j+1})$, due to the charge of the neighboring islands.

In Fig. 9 we plot the pumped charge $Q_\infty(\phi = 0)$ at the onset of the clipped regime ($\delta J = J_0$), as a function of both $E_L$ and $E_{CC}$. $Q_\infty$ is remarkably stable against the nearest neighbor interactions and there is a wide region of the phase diagram where the pumped charge is quantized. The role of $E_{CC}$ seems indeed to be negligible for the Rice-Mele pumping: although the Hamiltonian $H_{RM,\text{tot}}$ does not conserve the CP number, its many-body ground states are in good approximation half-filled states where CPs are localized on every other SC island for most of the driving protocol. Therefore, the role of the interaction is marginal. For realistic ranges of $E_{CC}$, our simulations indicate that the nearest-neighbor interactions do not cause any topological phase transition in the effective 2D Floquet topological insulating state that determines the onset of the quantization of the pumped charge.

## 4   Results of the Harper-Hofstadter model

The HH model contains two main differences from the RM model. First, its implementation does not require modulation of the tunneling amplitudes but only of the induced charges. This

makes it more suitable for experimental implementations but, at the same time, more susceptible to finite-size effects since this protocol does not rely on any dimerization. Second, when the Aharonov-Bohm phase $\Phi$ is set to $2\pi/3$, the model displays two insulating phases with opposite Chern numbers and filling $1/3$ and $2/3$.[1] This implies that by varying the parameter $\bar{\mu}$ in Eq. (13), it is possible to invert the direction of the pumped charge. The two insulating states correspond to intervals in $\bar{\mu}$ with bounds approximately given by the band gaps. These states are separated by a gapless phase in which pumping is not quantized. In addition, since the topological insulating Floquet states of the HH model appear at fillings $1/3$ and $2/3$, the role of nearest-neighbor interaction is less trivial than in the RM model where interactions favor, to a certain extent, the dimerization characterizing the half-filled topological state. In the hardcore boson limit, however, a more careful analysis shows that the two insulating phases of the HH model are related by a particle-hole-like symmetry which is fulfilled also in the presence of the nearest-neighbor interaction in Eq. (24). This symmetry is defined by

$$\hat{b}_r \to \hat{b}_r^\dagger, \tag{25}$$

$$n_{g,j} \to 1 - n_{g,j}. \tag{26}$$

The second equation corresponds to the mapping $\mu_{0,j} \to -\mu_{0,j}$ and $t \to t + \tau/2$. This symmetry relates states with complementary fillings and implies that the pumped charge changes sign when reflecting $n_{g,j}$ around $\frac{1}{2}$. In particular, this holds for any value $E_{CC}$, implying that the effect of the interaction is insensitive to the average charge density. To control the filling factor and, therefore, the direction of the current, we tune the offset of the chemical potential $\bar{\mu}$ through the average value of the induced charge during one driving period:

$$\bar{n}_g = C^g V_0 = \frac{1}{2} - \frac{\bar{\mu}}{2E_C}. \tag{27}$$

Here we consider, for simplicity, uniform $C^g$ and $V_0$ across the chain.

To investigate the stability of Thouless pumping in the presence of interactions, we consider the dynamics dictated by the Hamiltonian

$$\hat{H}_{HH,\text{tot}}(t,\phi) = \hat{H}_{HH}(t) + \hat{H}_b(\phi) + \hat{H}_{\text{int}}(t), \tag{28}$$

defined by the Hamiltonians in Eqs. (13), (15) and (24) with the drive obtained by a modulation of the induced charge with position-dependent phases $\chi_j = 2\pi j/3$. We calculate the charge using Eq. (21): the system is initialized in the ground state of the Hamiltonian in Eq. (28) at $t = 0$ and the time dependence of the Floquet states is computed by solving numerically the Schrödinger equation.

We first examine the pumping for values of $\bar{\mu}$ where the system lies deep in one of the Floquet topological insulator phases. Fig. 10 shows the behavior of $\bar{Q}$ as a function of the lead coupling $E_L$ for different values of the nearest-neighbor interactions, up to half the charging energy, $E_{CC} < E_C/2 = 2E_J$. The behavior is qualitatively analogous to that observed for the RM pumping with clipped modulation (compare Fig. 8(a) and Fig. 10): a good quantization of $\bar{Q}$ is achieved for sufficiently strong $E_L \gtrsim 0.4E_J$. Importantly, even strong interactions ($E_{CC} = 1.9E_J$ in the data) do not significantly alter the behavior of the pumped charge, which appears remarkably stable.

We emphasize that for the HH model, averaging over the phase difference $\phi$ is necessary to obtain the predicted quantization of the pumped charge since we consider small system sizes ($M = 6, 9$). This is caused by the winding of the eigenenergies of the populated Floquet states, displayed in Fig. 11. For the most occupied state at filling $1/3$ [panel (a)] the quasienergy

---

[1]The appearance of topological bands with nonzero Chern numbers is not limited to $\Phi = 2\pi/3$ but holds for any rational value of the flux $\Phi = 2\pi p/q$, with $p, q \in \mathbb{N}$, which results in up to $q$ non-overlapping bands.

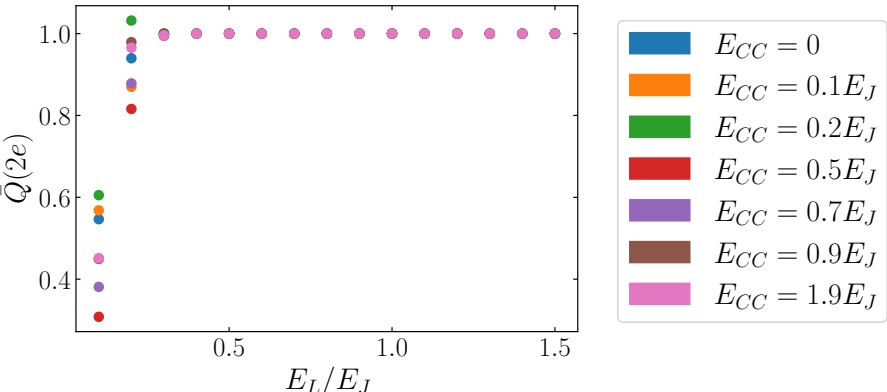

Figure 10: Pumped charge $\bar{Q}$ as a function of the coupling with the leads $E_L$ for different values of the nearest-neighbor interaction. The data are obtained with parameters $M = 6$, $E_C = 4E_J$, $\delta\mu = E_J$, and $\omega = 0.01E_J$, in the gapped topological phase at filling $1/3$.

winds three times in the positive direction and twice in the negative direction, resulting in a total winding number $+1$. Panel (b) in Fig. 11 shows instead the opposite winding for the gapped phase at filling $2/3$, corresponding to pumping in the opposite direction. In both cases, the current $\partial_\phi \mathcal{E}(\phi)$ strongly depends on $\phi$ and even changes sign (see Fig. 12(a) for the state at filling $1/3$). Therefore, a quantized pumping can be obtained only by averaging over $\phi$. This is a characteristic of small-size systems. By comparing the results for $M = 6$ and $M = 9$ in Fig. 12(a) we see that the phase dependence considerably decreases with the chain length. However, in general, the HH model is considerably more influenced by finite-size effects than the RM model in the clipped regime (see Fig. 5 for comparison).

In an experimental setup, control over the voltage offset $V_0$ of the side gates provides a possibility of interpolating between the two gapped phases and exploring the phase diagram of the system as a function of $\bar{\mu}$ (Fig. 12(b)): our results clearly show the appearance of the two topological Floquet phases with quantized charge $\bar{Q} = \pm 1 \cdot 2e$ both without interactions, $E_{CC} = 0$, and for strong interactions, $E_{CC} = E_J/2$. A large value of $E_{CC}$ seems indeed to be beneficial for the stability of the topological phases, as it increases the width of the plateaus where the pumped charge is quantized.

In the data of Fig. 12(b), the symmetry in Eqs. (25) and (26) is not exactly reflected. The discrepancy results from the fact that the initial time of the pumping protocol is the same for all values of $\bar{\mu}$. The missing translation $t \to t + \tau/2$ violates the requirement (26), but the effect is only seen when $\bar{\mu}$ is close to the band extrema and the initial conditions strongly affect the pumping outcome. This strong dependence is shown explicitly in the inset of Fig. 12(b), where $\bar{Q}$ fluctuates with $\bar{n}_g$ around the transition between the gapless metallic states ($\bar{n}_g \lesssim 0.35$) and the Floquet topological insulator, ($\bar{n}_g \gtrsim 0.35$). Indeed, when $\bar{\mu}$ lies close to a topological band edge, a small change in the SC phase difference $\phi$ can change the initial state between being either metallic or insulating. Consequently, $Q_\infty(\phi)$ has discontinuities in $\phi$ and its phase average $\bar{Q}$ strongly depends on the driving frequency, the precise sampling of $\phi$, and the system size, which is reflected in the single-particle level spacing within the bands.

We show this irregular behavior in Fig. 12(a), where we plot the infinite-time-averaged pumped charge $Q_\infty$ as a function of the SC phase $\phi$. A comparison of the data obtained at the edge ($\bar{n}_g = 0.35$, orange triangles) and inside ($\bar{n}_g = 0.4$, teal circles) the topological phase clearly shows the different role of $\phi$ in the two cases. While the amplitude of the variation of $Q_\infty$ is the same, in the topological phase the pumped charge has a smooth dependence on $\phi$ and it oscillates around the quantized value $\bar{Q} = 1$ marked by the horizontal dashed line.

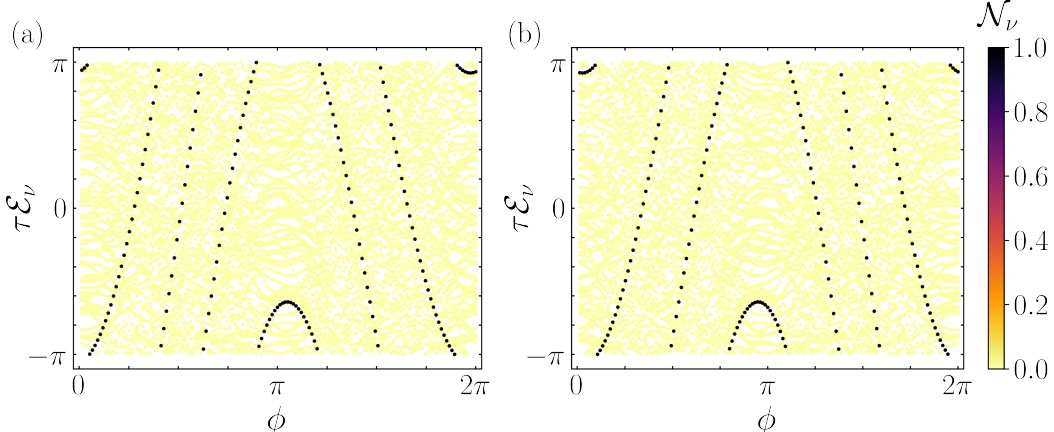

Figure 11: Winding of the quasienergies of the many-body populated state as a function of the lead phase difference $\phi$. (a) Winding in the gapped phase at filling 1/3 ($\bar{n}_g = 0.4$). We observe that the quasienergy winds three times in the positive direction and twice in the negative direction, corresponding to a Chern number $C_1 = 1$ of the lowest band. (b) Winding in the phase at filling 2/3 ($\bar{n}_g = 0.6$). The winding is opposite with respect to panel (a).

On the edge between the metallic and the insulating phase, on the other hand, $Q_\infty$ has many discontinuities that are reflected in the fluctuations of $\bar{Q}$ in the inset of panel (a).

## 5 Experimental implementation of the pumping schemes

The analysis presented in the previous sections and the observation of a quantized Thouless pumping rely on several constraints, important in devising an experimental realization of the presented models.

The most fundamental parameter in both driving protocols is the frequency $\Omega$. On one side, $\Omega$ determines the ideal pumped current $I = 2e\Omega\mathcal{C}$; therefore, in order to obtain clearly measurable currents ($I \gtrsim 10\text{pA}$), we require $\Omega \gtrsim 300\text{MHz}$. On the other, the frequency determines the onset of nonadiabatic errors, typically scaling as $(h\Omega/E_g)^2$. Therefore $\Omega$ must be sufficiently small compared to the many-body gaps $E_g$, although the presence of dissipation might mitigate this requirement by suppressing some of the nonadiabatic corrections [56]. For both the RM model with clipped modulation and the HH model at $\Phi = 2\pi/3$, the gap is given by the minimum between $2\delta\mu$ and a Josephson term proportional to $E_J$ (for sufficiently large $E_L$). $\delta\mu$ is proportional to $E_C$, and to enter the regime of quantized pumping it is necessary that $E_C \gtrsim E_J$. $E_J$ can be tuned more easily than $E_C$, for example by applying a suitable voltage to a global gate as in the devices analyzed in Ref. [10]. Therefore, we consider the charging energy as the limiting factor in determining the gap which, for small superconducting islands, is approximately of the order of $h30$ GHz [47]. This poses an upper limit to the frequency, $\Omega \lesssim 10$ GHz, to avoid excessive nonadiabatic errors. The frequency range $\Omega \in (300 \text{ MHz}, 10 \text{ GHz})$ of the voltage drive of the electrostatic gates can be explored with standard waveform generators.

The RM model has the disadvantage of requiring modulation of both the induced charges and the Josephson couplings. This can be particularly challenging as the cutter gates controlling the Josephson energies may additionally induce charge on the neighboring islands. Such an effect can, however, be compensated with suitable corrections of the voltages $V_g$ of the side gates. At the same time, the RM model displays, for $M = 6$ islands, smaller finite-size effects

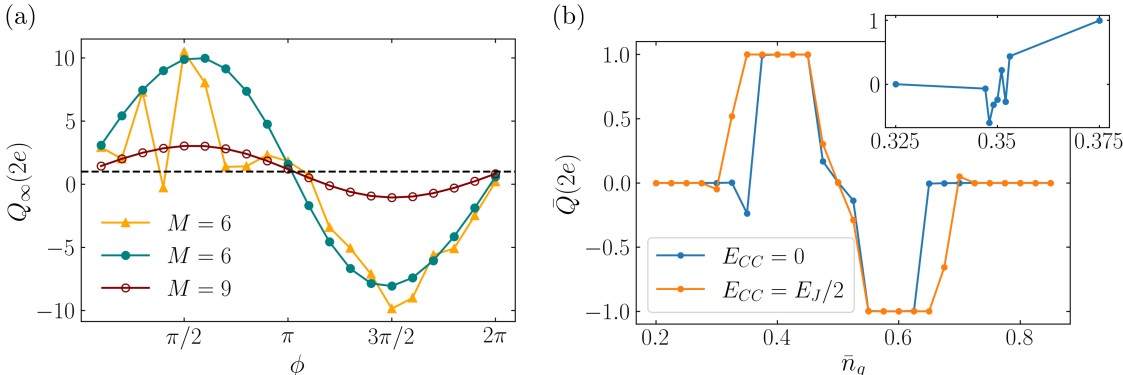

Figure 12: Pumped charge in the hardcore boson HH model. (a) Dependence of the pumped charge on the phase difference $\phi$ for chains of $M = 6$ and $M = 9$ islands around the transition $\bar{n}_g = 0.35$ (orange) and in the topological insulating phase $\bar{n}_g = 0.4$ (teal and maroon). (b) Averaged pumped charge $\bar{Q}$ for $M = 6$ islands. The blue curve corresponds to vanishing nearest-neighbor interactions. The orange curve is obtained for $E_{CC} = E_J/2$. Interactions extend the Floquet topological phases. Inset: fluctuations of $\bar{Q}$ close to the transition between the metallic and the topological insulating phases for $E_{CC} = 0$.

than the HH model, and, in particular, does not show a strong dependence of the pumped charge $Q_\infty$ on the lead phase $\phi$, especially in the clipped regime. For the same number of superconducting islands, the HH model requires control of only half of the electrostatic gates but displays instead a very strong dependence on the phase $\phi$ for $M = 6$ [Fig. 12(a)]. To avoid this limitation, two strategies can be envisioned: either implementing the pumping in longer chains, or embedding the system in a device that allows for averaging in time over the phase $\phi$, similar to the proposals of Refs. [50, 57].

Concerning the scalability of the HH model at flux $2\pi/q$, only $q$ independent voltage signals are needed for SC chains sufficiently uniform, such that the capacitances $C_j^g$ and charging energies present only minor variations along the system. These $q$ signals need then to be distributed across suitable gate structures.

Regarding the averaging over the phase $\phi$, several options can be envisioned. The first possibility is based on introducing a suitable voltage bias $V_b$ between the two external SC leads (see Ref. [50]). This yields a linear winding of the phase $\phi$ with period $\tau_\phi = h/2eV_b$. The number of pumped CPs in this period is given by $Q(\tau_\phi) = \mathcal{C}\Omega\tau_\phi$, and the average over the phase $\phi$ is suitably implemented if $\Omega\tau_\phi \gg 1$. An alternative method would require instead to embed the HH Josephson junction chain in a superconducting ring, in order to impose a phase bias $\phi$ that can be varied in time through a driven magnetic flux.

In our analysis of short Josephson junction chains, we did not consider explicitly the role of disorder. For both the RM Hamiltonian (9) and the HH Hamiltonian (13), the onsite disorder corresponds to a position dependence of the time-independent part of $\mu_j$ in Eq. (8). This can be caused by non-uniform capacitances $C_j^g$ and a failure in balancing them with the voltages $V_{0,j}$. The effect of this kind of disorder on Thouless pumping has been extensively studied (see, for instance, Refs. [34–37]); in general, the quantization of the pumped charge is robust as long as the random variations of the onsite potential are weaker than the energy gap. In Coulomb-blockaded JJAs, the disorder amplitude of the onsite energy depends on the variance of $E_C\bar{n}_g$, whereas the energy gaps are determined by the Josephson energies. Therefore, to approach an accurate quantization of the pumped charge, we need to consider a balance between the following competing constraints.

On one side, the ratio $E_C/E_J$ cannot be too large. Specifically, the standard deviation of the island-dependent $E_C \bar{n}_g$ (where $\bar{n}_g$ represents the targeted $n_g$ average) must be considerably smaller than $\delta J$ and $E_J$ in the RM and HH pumping schemes, respectively. Our calculations rely on the CP hardcore assumption $E_C \gg E_J$; however, numerical investigations [39] of the RM model with interactions of the form (2) reveal that the RM Floquet topological insulator phase survives even when the ratio $E_C/\delta J$ decreases to $E_C/\delta J \sim 3$ if $\delta \mu$ is sufficiently strong. Therefore, keeping a moderate value of $E_C/E_J \gtrsim 3$ may be beneficial to reduce onsite disorder and nonadiabatic effects. On the other side, the Josephson energies cannot exceed the threshold corresponding to the insulator-SC transition in the static case. Thouless pumping can indeed be realized only when the instantaneous energy spectrum of the system and the related ground states at each time $t$ correspond to insulating phases.

A different kind of disorder characterizing JJAs are random variations in the Josephson energies and, in the RM model, differences in the functions $E_{J,j}(V_{c,j})$ associated with the junctions along the chain. Also in this case, the related disorder in the hopping terms of Eq. (6) becomes detrimental for an accurate quantization of Thouless pumping when the amplitude becomes comparable with the energy gaps of the systems.

Given the accuracy of the lithographic techniques adopted for the fabrication of hybrid JJAs, however, we expect the typical disorder amplitudes in the Josephson energies to be below 10% (see related experimental estimates in Ref. [58]). A disorder strength in this range is not harmful for the implementation of Thouless pumping as it is way below the necessary threshold to close the many-body gap and suppress quantized transport.

# 6   Conclusion

Motivated by recent advances in fabrication techniques, we numerically investigated possible driving protocols to implement topological Thouless pumping in short 1D arrays of tunable Josephson junctions. We considered the JJAs to be in a Coulomb-blockaded regime, where the charging energy of each SC island and the SC gap are the dominant energy scales, allowing us to approximate CPs as hardcore bosons. To study quantized transport in the long-time regime, we connected the array to two grounded SC leads, breaking the conservation of particle number. We used Floquet theory to extract the pumped charge at a small but finite driving frequency in the limit of an infinite number of driving cycles.

We focused on two prototypical models for topological quantum pumping, the periodically driven Rice-Mele and Harper-Hofstadter models. For both, we analyzed the role played by the coupling with the SC leads and their phase bias, as well as the effect of nearest-neighbor interactions originating from cross-capacitance between the SC islands. These ingredients are specific to the solid-state implementation of Thouless pumping with JJAs, and they extend the recent analyses of the role of interactions in Floquet topological insulators inspired by ultracold-atom experiments [23, 27, 39]. Furthermore, their understanding is essential for a successful experimental realization of topological quantized transport in hybrid SC-SM devices. Both models display remarkable robustness with respect to nearest-neighbor interaction which does not affect the transport properties, even when it becomes larger than the Josephson tunneling. Moreover, the coupling to the leads helps to stabilize transport as it gaps out the zero-energy edge modes, which would otherwise appear in an open chain, and allows for an adiabatic evolution at sufficiently low driving frequency.

In the RM model, finite-size effects are most easily suppressed in a clipped driving regime where the modulation of the Josephson coupling is tuned such that the SM substrate below the Josephson junctions is depleted at half periods, effectively dimerizing the chain. However, the implementation requires simultaneous tuning of both the Josephson couplings and the charge

induced on each island, resulting in a more complicated experimental protocol.

The HH model, on the other hand, requires fewer control gates as the Josephson couplings are constant in time. The drawback is that for small system sizes ($M = 6, 9$ islands) we find that it is necessary to dynamically average over the phase difference of the external leads to obtain a good quantization of the pumped charge. This behavior appears analogous to previous proposals to achieve quantized pumping in short Josephson junction chains [50]. However, even at constant phase bias, the discrepancies from the ideal quantized case rapidly decrease with the system size. Moreover, the HH model displays a richer topological phase diagram since the number of bands with nontrivial Chern number depends on the effective magnetic flux $\Phi$ which in turn is determined by the position-dependence of the on-site energy modulation. In the simplest scenario, where $\Phi = 2\pi/3$, the tuning of the average induced charge $\bar{n}_g$ on the SC islands allows for control of the chemical potential. Tuning this to lie in different band gaps leads to a quantized current flowing in different directions.

Our results suggest that quantized Thouless pumping is experimentally accessible with the recently developed SC-SM JJAs, paving the way for a new generation of experiments investigating topological transport in Floquet systems as an alternative to ultracold-atom approaches. However, there are several further effects worth investigating to improve our analysis. First, one could relax the hardcore boson approximation and allow for a standard onsite repulsion, which has been studied for the RM model with periodic boundaries [38–40, 59]. We expect that an intermediate ratio $E_C/E_J \gtrsim 3$ is beneficial for experimental realizations because it would mitigate the effects of disorder in the induced charges without driving the system away from its insulating phases. In this situation, our estimates indicate that driving frequencies $\Omega \gtrsim 300$MHz result in currents $I \gtrsim 10$pA without introducing excessive non-adiabatic effects. Another important element to consider is the possible presence of longer-range interactions. Indeed, the inverse of the capacitance matrix is approximately tri-diagonal only if the self-capacitance of the SC islands is much larger than their cross-capacitance. While this can be a reasonable approximation, depending on the geometry of the JJA, it is important to examine the pumping robustness when it breaks down. Far from being necessarily a problem, longer-range electrostatic interactions can be exploited to study the transport of fractions of CPs pumped at each cycle [57]. Finally, solid-state devices are subject to many decoherence channels, either due to quasiparticle poisoning, incoherent tunneling, charge noise, or scattering from crystal defects and imperfect interfaces. While precise modeling of these effects is prohibitive, an interesting future perspective is investigating the robustness of quantized transport with respect to simpler decoherence mechanisms, such as local dissipation or dephasing, similarly to Ref. [56], or incoherent coupling between the external (superconducting or normal) leads and the SC islands of the JJA.

## Acknowledgements

We thank L. F. Banszerus, C. M. Marcus and S. Vaitiekėnas for useful discussions.

**Funding information** This project was supported by the Villum Foundation (Research Grant No. 25310) and received funding from the European Union's Horizon 2020 research and innovation program under the Marie Skłodowska-Curie Grant Agreement No. 847523 "INTERACTIONS."

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
