# Peer review of "Thouless pumping in Josephson junction arrays"

_SciPost Physics_

## Round 1 · Referee Report · Anonymous (Referee 1) · 2023-11-17

Report

The authors study short hard-core boson chains ( Rice-Mele and the Harper-Hofstadter models), driven by time periodic perturbations and focus on Thouless pumping. In my opinion, this paper is better suited to Scipost Physics Core as it does not satisfy the general acceptance criteria of Scipost Physics. Most of the paper deals with non-interacting but driven short chains, which is correct but only moderately interesting.

Later on, interaction effects are also discussed briefly. However, given the shortness of the chain (6-9 sites), the level spacing is large and the considered repulsive interaction at most probably open up a gap in the spectrum which is still negligible compared to level spacing. It would be interesting to crank up the interaction further as well as quantify the expected Mott gap.

Is it possible to engineer attractive interaction for these hard-core bosons? If yes, it would be fruitful to investigate this as well since the dominant instability is phase separation, and its effect should be distinct from repulsive interactions.

Also while short chains seem to be relevant experimentally at the moment, it would make sense to look into longer chains as well to minimize finite size effects or to be able to determine to what extend the finite system size affects their results.

Do the authors use many-body exact diagonalization to study the interacting model? A minimal description of the method would be great.

What are the model parameters for Fig. 11?

  • validity: good
  • significance: ok
  • originality: good
  • clarity: top
  • formatting: perfect
  • grammar: perfect

Author:  Michele Burrello  on 2023-12-15  [id 4195]

(in reply to Report 1 on 2023-11-17)

We thank the Referee for the attentive reading of our work. We carefully considered the comments presented by the Referee in the revision of our manuscript, and we address them in detail below.

The authors study short hard-core boson chains ( Rice-Mele and the Harper-Hofstadter models), driven by time periodic perturbations and focus on Thouless pumping. In my opinion, this paper is better suited to Scipost Physics Core as it does not satisfy the general acceptance criteria of Scipost Physics. Most of the paper deals with non-interacting but driven short chains, which is correct but only moderately interesting.

We respectfully disagree with the Referee concerning the general acceptance criteria and interest of our work. We emphasize indeed that this is the first work proposing electrically tunable Josephson junction arrays as a platform for quantum simulations and, specifically, for the realization of Floquet states out of equilibrium. In Sec. 2, for instance, we set the stage for a general modelling of these tunable devices which will be useful for future works. Furthermore, this is also the first work, to the best of our knowledge, addressing nearest-neighbor interactions in the Rice-Mele model. Indeed, on this subject, we found only arXiv:2308.13375, which was indicated by Referee B and appeared on the arxiv after our manuscript.

It seems to us that there is considerable attention towards the effects of interactions in pumping schemes, as testified, for instance, by Refs. [23,39-41,58,62] of the revised manuscript. The analysis of interactions is a crucial part of our work and, to emphasize further its effects, in this revision, we extended Sec. 3.3, Sec. 4 and Sec. 5, and we added an appendix on the breakdown of the hardcore approximation. In the following, we address the specific points raised by the Referee.

Later on, interaction effects are also discussed briefly. However, given the shortness of the chain (6-9 sites), the level spacing is large and the considered repulsive interaction at most probably open up a gap in the spectrum which is still negligible compared to level spacing. It would be interesting to crank up the interaction further as well as quantify the expected Mott gap.

We are surprised that the Referee writes that interactions effects are "discussed briefly": Sec. 3.3 and most of Sec. 4 are dedicated to the effects of nearest-neighbor interactions and, in particular, Figs. 9, 10 and 12(b) are meant to show the robustness of our pumping schemes against interactions, whereas Fig. 11 refers to the many-body quasienergies in the presence of weak interactions. We updated these sections and the caption of Fig. 11 to make these aspects clearer. Additionally, further details on the onsite interactions are now addressed in Appendix A.

Concerning the magnitude of the interactions, for instance, Fig. 9 displays data with interactions up to $E_{CC}=3J_0$. As a comparison, the single-particle level spacing for $M=6$ is about $J_0/2$. Additionally, in the system with open boundaries and coupled to the superconducting leads, the relevant many-body gap is set by $E_L$ when $E_L < J_0$. Therefore, when $E_{CC} \gtrsim J_0$, interactions are definitely not negligible. What we observe, instead, is that the Rice-Mele pumping scheme is incredibly robust against nearest-neighbor repulsions.

We stress that our simulations refer to hardcore bosons (with the exception of the new data presented in appendix). This implies that, for generic time and for both pumping schemes, the instantaneous Hamiltonians are fully in a Mott insulating phase (corresponding to a Coulomb valley regime for each SC island). The data in appendix suggest that this assumption breaks when $E_C$ decreases below $2J_0$ in the Rice-Mele protocol.

Concerning the nearest-neighbor interactions in the Rice-Mele model, they favour the formation of charge density waves with occupations $010101$ or $101010$ due to the staggered induced charges. These states, however, are not detrimental for the RM pumping, since the modulation of the induced charges let them evolve one into the other. Strong interactions, instead, affect the HH in a stronger way and, in Sec. 4, we added a comment about the fact that for $E_{CC} \gtrsim 2E_J$ the HH pumping vanishes. In general, we revised Secs. 3.3 and 4 to make these aspects clearer.

Is it possible to engineer attractive interaction for these hard-core bosons? If yes, it would be fruitful to investigate this as well since the dominant instability is phase separation, and its effect should be distinct from repulsive interactions.

The electrostatic term $E_{CC}$ is repulsive and, in the RM model in which $n_g$ alternates between the islands with values larger and greater than 0.5 ($\mu$ changes sign), it favors the onset of the above mentioned density waves rather than Mott states $000000$ and $111111$. These Mott phases would appear shifting the average chemical potential $\mu_0$ away from 0, thus having the islands all in the same Coulomb valley. It is important, however, to observe that the induced charges $n_g$, which depend on time and position, enter the expression of the nearest-neighbor repulsions in Eqs. 2 and 24. This implies, for instance, that in the HH model the effect of the interactions is less trivial since the half-filled charge density waves compete with the running density waves of the kind $100100$ [or their particle-hole conjugate], which may even be favored by the interactions for certain choices of the pumping parameters. The effect of moderate interactions on the HH protocol is the one explicitly shown in Fig. 12(b): overall, interactions of magnitude $E_{CC}\sim E_J/2$ enlarge the topological pumping regimes. When $E_{CC}$ grows above a certain threshold, however, the system undergoes a transition towards half filled density waves which are not compatible with the HH pumping. Also when the average chemical potential is misplaced or the modulation of the induced charges is too small, the system is blockaded and transport blocked. We addressed these aspects in the revised Sec. 3.3 and 4.

Finally, our system does not suffer from phase separation because, given the external leads, the particle number is not conserved and the filling adjusts to the most energetically convenient many-body state. In this respect it is very different from closed ultracold atom systems.

Also while short chains seem to be relevant experimentally at the moment, it would make sense to look into longer chains as well to minimize finite size effects or to be able to determine to what extend the finite system size affects their results.

Finite size effects on Thouless pumping in noninteracting chain is well known in the literature and one of us has analyzed them in detail in his MSc thesis. The rapid decay of finite size effects can also be seen from Fig.5, where the data for $L=8$ already hardly display any oscillations with $\phi$ at all. In that case, the overall corrections to quantization (diabatic errors + finite size effects) is less than 0.01. Fig. 12(a) provides a similar comparison for the HH model, which is quantitatively more affected by finite-size effects.

Do the authors use many-body exact diagonalization to study the interacting model? A minimal description of the method would be great.

Yes, we use exact diagonalization using the qutip python package and construct explicitly the Floquet operator to compute the time evolution at arbitrarily long times. We made this more explicit in the text.

What are the model parameters for Fig. 11?

The parameters are $M=6$, $E_C=4E_J$, $E_{CC}=0.2E_J$, $\delta_\mu=E_L=E_J$ and $\omega=0.01E_J$. We added them in the caption.

---

## Round 1 · Referee Report · Anonymous (Referee 2) · 2023-11-20

Strengths

1- Very creative idea, very timely.
2- Very good discussion of how model parameters relate to JJ junction parameters
3- Very good discussion of limiting factors/realistic aspects of the set up
4- Promising proposal for a new avenue for quantized pumping
5- Very good account of related literature, comprehensive discussion

Weaknesses

1- Some additional simulations showing robustness of the pumping with respect to parameters other than driving frequency would be good
2- Advantages/ Disadvantages compared to quantum simulator realizations could be emphasized more.

Report

Thouless charge pumps have recently been attracting attention due to their
successful realization in quantum simulators, opening up an avenue for
investigating topology in slowly driven periodic processes as a one+one
dimensional relative of the quantum Hall effect.

In solid state systems, the realization of quantized pumping is hampered by several aspects: the difficulty of driving onsite potentials and tunneling matrix elements independently and coherently, the presence of dissipation channels and complication of a open system.

The present manuscript proposes to work with arrays of Josephson junction
arrays where electrical tuning of gates is suggested to provide the
necessary means of modulating the system in time. The authors provide
a discussion of the physics of such system and then limit the analysis
to the hardcore boson limit. They study two limiting case, the Rice-Mele
regime and the Harper-Hofstadter regime and related the model parameters
to those of the underlying Josephson-Junction arrays.

Implementing Thouless pumping in an additional experimental platform
would be very exciting and the paper comes up with original and creative
ideas. This work is very timely and will certainly be of interest to scientists
working on topological systems as well as experimentalists interested
in realizing quantized transport. This manuscript should therefore
be published in SciPost after the authors gave consideration to the following remarks.

Requested changes

1) The manuscript could emphasize more which aspects of pumping can be studied in the JJAs that can't be done with ultracold atoms. The authors rightfully stress that in JJAs, they would be dealing with charged particles, but perhaps a stronger points can be made. Some aspects are touch upon at the end the manuscript, yet this question should be given higher priority in the text.

2) The robustness of quantized pumping is the ultimate question.
The authors mention some aspects throughout but they should ideally propose a number of parameters that one could vary to see robust quantized pumping. They discuss the driving frequency, what else could be tuned? Ideally, some example could be added.

3) Is it possible to study topological transitions in this system? That would be very exciting.

4) The proposed setup would realize an open system (i.e., embedded into an environment), taken into account via the boundary term from Eq. 15. Why is quantization guaranteed? Is the quantization limited to specific boundary terms? This general question -- quantization in open systems with environment -- should ideally be discussed more in the text
(beyond the Floquet analysis).

5) The system may realize nearest neighbor interactions. Wouldn't these mess
with the sublattice symmetry in the Rice-Mele model case? That is, the protecting symmetry of the SSH model would be lost.

6) As a remark, nearest neighbor interactions were realized in quantum gas experiments in Ferlaino's group, Innsbruck. A recent theory paper discusses Thouless pumping for that case, arXiv:2308.13375.

7) Is there any limitation on the number of pump cycles, due to e.g., dissipation, decoherence, defects,...? If many cycles are possible, then this would be a great advantage over current quantum gas realizations. Why is dissipation and the coupling to the environment not a major issue? What about phonons?

8) The authors could also cite Nakajima et al that experimentally studies
Thouless pumping in the presence of disorder.

9) Generally, repeating some earlier comments, the paper could be stronger if it was more clearly demonstrated where quantized pumping will break down away from ideal parameters (expressed in terms of JJ physics), supported by numerical simulations.

  • validity: high
  • significance: top
  • originality: top
  • clarity: high
  • formatting: perfect
  • grammar: perfect

Author:  Michele Burrello  on 2023-12-15  [id 4194]

(in reply to Report 2 on 2023-11-20)

We thank the Referee for appreciating our work and emphasizing its timeliness and general interest. Below we address the raised points.

1) The manuscript could emphasize more which aspects of pumping can be studied in the JJAs that can't be done with ultracold atoms. The authors rightfully stress that in JJAs, they would be dealing with charged particles, but perhaps a stronger points can be made. Some aspects are touch upon at the end the manuscript, yet this question should be given higher priority in the text.

In the updated manuscript we added new comments on the role of interactions compared with ultracold atoms at the beginning of Sec. 3.3 and, importantly, we added a discussion on the role of the external thermal bath at the end of Sec. 4. In this context we stressed that the typical dissipative effects due to cold environments in superconducting devices make it feasible to observe pumping in the steady Floquet regime (as in the case of the non-topological pumping achieved in the experiments in Refs. 33 and 34). This is a major difference between JJ chains and ultracold atom setups. The latter are indeed much more isolated and subject to heating, which limits the number of pumping cycles. We also added new references to the ultracold atom literature and additional comments in the conclusions.

2) The robustness of quantized pumping is the ultimate question. The authors mention some aspects throughout but they should ideally propose a number of parameters that one could vary to see robust quantized pumping. They discuss the driving frequency, what else could be tuned? Ideally, some example could be added.

We investigated the robustness against driving frequency, coupling with the external leads and cross-capacitance interactions. Concerning the HH model, we also considered the behavior as a function of the average induced charge in the system. In the revised version, we commented in more detail the breakdown of the HH pumping at high values of the nearest-neighbor interaction. We also added an appendix about the breakdown of the hardcore approximation: our data suggest that quantized pumping is stable for charging energies above a threshold of the order $E_C \gtrsim 2J_0$. A precise analysis of the regime at low charging energies, however, becomes particularly demanding on the computational level because it would require considering much larger Hilbert spaces.

3) Is it possible to study topological transitions in this system? That would be very exciting.

An example of a topological phase transition is already shown in Fig.12 (b), where the average value of the induced charge $n_g$ determines the filling and thus the Chern number of the ground state. In this case the phase transitions interpolate between gapped topological phases and gapless phases, therefore there is not a sharp topological phase transition between two gapped phases. In principle, it is also possible to study a topological phase transition in the Rice-Mele model, if one sets the Josephson couplings in such a way that the system never enters the topological phase of the SSH model (for instance by introducing a static staggering of the even and odd Josephson energies $E_{J,j}$). We don't see any fundamental experimental hindrance to tune the JJ array from one regime to the other.

4) The proposed setup would realize an open system (i.e., embedded into an environment), taken into account via the boundary term from Eq. 15. Why is quantization guaranteed? Is the quantization limited to specific boundary terms? This general question -- quantization in open systems with environment -- should ideally be discussed more in the text (beyond the Floquet analysis).

Concerning the external superconducting leads, our modelling of a coherent coupling stems from the fact that Cooper pairs form a condensate in the leads. Therefore this is a very special kind of environment and we emphasized further this aspect at the end of Sec. 2. Quantization is, in general, not guaranteed as shown, for instance in Fig. 8 and discussed in Sec. 3.2 (a weak coupling with the leads enhances non-adiabatic effects). A separate analysis would be required for a more general incoherent coupling with external leads which does not preserve the total number of particle (as in the case of normal metallic leads). In our opinion, superconducting leads treated as coherent (non-thermal) baths of Cooper pairs are the most natural and effective choice for this setup and the modelling is consistent with experiments in SC devices (see for instance Refs. 33 and 34) and the theoretical study of SC circuits. Normal metallic leads will complicate the picture and we left their analysis for future studies.

Concerning a more general thermal environment (as in the case of a phononic bath), instead, we considerably extended our discussion in Sec. 5 and in the conclusions. Essentially, if the system is coupled to a cold dissipative environment with a temperature sufficiently smaller than the relevant excitations, as expected for an array of JJ operating in a dilution fridge, the system will be projected in the time-dependent ground state, thus improving quantized pumping. This has been studied for instance in Ref. [59].

5) The system may realize nearest neighbor interactions. Wouldn't these mess with the sublattice symmetry in the Rice-Mele model case? That is, the protecting symmetry of the SSH model would be lost.

Yes, the chiral symmetry is lifted. However, the topological protection of the pumping does not rely on it, but rather on the topological nature of the 2D bands in (k,t) space of the model. In the case of the RM model, the universality class is the same of the Haldane honeycomb model, which breaks all symmetries. And, clearly, also in the HH model the universality class is A, like quantum Hall. The introduction of the nearest-neighbor interactions is non-trivial, also because, in the analog 2D picture, it amounts to non-local interactions along the 'frequency' direction. Our data on the quantized pumping in the presence of the nearest-neighbor interaction (Fig. 9) can be considered as an indicator of the underlying many-body Chern number and provide an estimate of the degree of resilience of these Floquet topological phases.

6) As a remark, nearest neighbor interactions were realized in quantum gas experiments in Ferlaino's group, Innsbruck. A recent theory paper discusses Thouless pumping for that case, arXiv:2308.13375.

We thank the Referee for this reference. We added a citation and an additional comment about nearest-neighbor interactions in ultracold atom setups at the beginning of Sec. 3.3.

7) Is there any limitation on the number of pump cycles, due to e.g., dissipation, decoherence, defects,...? If many cycles are possible, then this would be a great advantage over current quantum gas realizations. Why is dissipation and the coupling to the environment not a major issue? What about phonons?

As mentioned above, for the standard gaps and temperatures in typical experiments involving SC devices, heating is not considered a major issue. This is experimentally proved, for instance, by the (non-topological) pumping experiments in Refs. 33 and 34. Current measurements in these setups always refer to the non-equilibrium steady states of these systems and, differently from ultracold atoms, it is very hard to acquire information about transient states and a time resolution that allows us to observe single pumping cycles. Phonons and other thermal baths are typically characterized by temperatures in the range 20 - 100 mK, thus one order of magnitude lower than the relevant many-body gaps we expect based on charging and Josephson energies. This allows us to dissipate heat, rather than heating the system as in ultracold atom setups which are much more isolated. We added several comments about these aspects in Sec. 5 and the conclusions.

8) The authors could also cite Nakajima et al that experimentally studies Thouless pumping in the presence of disorder.

We thank the Referee for the advice. We added a comment and a reference to Nakajima et al., Nat. Phys. 17, 844 (2021).

9) Generally, repeating some earlier comments, the paper could be stronger if it was more clearly demonstrated where quantized pumping will break down away from ideal parameters (expressed in terms of JJ physics), supported by numerical simulations.

Concerning the breakdown of the pumping, we added some comments in Sec. 5 and additional data in appendix about the breakdown of the hardcore limit. In particular, we considered the exact diagonalization of the Rice-Mele system by including 4 charge states per island. Our data suggest that quantized pumping can be retrieved for $E_C > 2J_0$, consistently with former results in the literature (Ref. [40]).

---

## Editorial Decision

resubmitted